# The Risk of the Aggravation of Diabetic Foot According to Air Quality Factors in the Republic of Korea: A Nationwide Population-Based Study

**DOI:** 10.3390/ijerph21060775

**Published:** 2024-06-14

**Authors:** Saintpee Kim, Sungho Won, Young Yi

**Affiliations:** 1Department of Orthopedic Surgery, Gangbuk Etteum Hospital, Seoul 01170, Republic of Korea; ksb8060@gmail.com; 2RexSoft Corps, Seoul 08826, Republic of Korea; won1@snu.ac.kr; 3Graduate School of Public Health, Seoul National University, Seoul 08826, Republic of Korea; 4Department of Orthopedic Surgery Foot and Ankle Service, Inje University Paik Central Hospital, Seoul 04551, Republic of Korea

**Keywords:** diabetic foot, lower extremity amputation, air quality, diabetic vasculopathy, diabetic neuropathy

## Abstract

This study aims to examine the association between the occurrence of diabetic foot and air quality (SO_2_, CO, NO_2_, O_3_). Open data were collected to conduct a big data study. Patient information was gathered from the National Health Insurance Service, and the National Institute of Environmental Science’s air quality data were used. A total study population of 347,543 cases were reviewed (case = 13,353, control = 334,190). The lag period from air quality changes to the actual amputation operation was calculated for each factor. The frequency of diabetic foot amputation in each region was identified and analyzed using a distributed lag non-linear model. Gangwon-do showed the highest relative risks (RRs) for SO_2_ and CO, while Chungcheongnam-do exhibited the highest RR for NO_2_. Jeju had the highest RR for O_3_. Regions like Incheon, Busan, and the capital region also showed significant risk increases. These findings emphasize the importance of tailored air quality management to address diabetic foot complications effectively.

## 1. Introduction

The burden of chronic disease has been increasing over the past several decades as life expectancy has increased, and the number of people with diabetes has increased exponentially [1]. Diabetic foot is a typical complication of patients with diabetes in the area of foot ankle orthopedic surgery. Diabetic foot refers to the comprehensive case of sensory changes and blood flow abnormalities in the foot associated with diabetic neuropathy and diabetic vasculopathy, from which ulcers and wounds occur [2]. In particular, it shows a characteristic of not healing easily if foot wounds occur as a result of decreased white blood cell function due to hyperglycemia or ischemia and malnutrition due to impaired circulation. For the above reasons, diabetic feet are not easily treatable, wound recovery is often delayed, and there is a high risk of infection and necrosis leading to atraumatic lower limb amputation.

It was reported that approximately 15 to 25% of people experience diabetic foot at least once in their lifetime [3]. Foot ulcers account for around 85% of the causes of amputations performed on patients with diabetes [3,4]. The prevalence of diabetic foot ulcers and diabetic foot wounds that require surgical treatment is reported to be 2–5% of the total population. Diabetic foot ulcers increase the five-year mortality risk by 2.5 times, and the five-year mortality rate of diabetes-related amputation is over 70%. A study that reported a five-year survival rate of 50–60% for diabetic foot ulcers also reported that diabetic foot ulcers showed worse outcomes than the survival rate for common cancers [5].

According to a previous study on diabetic foot amputation in Korea, the number of patients with diabetic foot and the number of procedures have been continuously increasing every year, and the related social and economic costs are steadily increasing. Preventing the aggravation of diabetic foot and accurately predicting how often it occurs is of the utmost importance in various areas, including public health, the socioeconomic field, and academia.

A dynamic analysis of diabetic foot and analysis of major aggravating factors are fundamental. The aggravating factors can be categorized into personal, socioeconomic, and environmental factors. In Korea, analytical studies on personal and socioeconomic factors are actively conducted to prevent diabetic foot [5], whereas a systematic analysis and big data studies on environmental factors are not conducted as actively.

It has also been reported that the condition of air quality contributes to the occurrence and exacerbation of diabetes as a major external factor [6,7,8]. The decline in air quality increases the impairment of glucose tolerance in patients with diabetes and degrades the function of the lining of blood vessels through oxidation. However, most studies so far have analyzed the correlation between the decline in air quality and the exacerbation or occurrence of diabetes [6,7,8]. There have been reports of the correlation between air quality and the worsening of complications [9], but there are no reports of the aggravation of diabetic foot due to changes in air quality. Accordingly, it is necessary to conduct studies related to wound deterioration and amputation due to changes in air quality in patients with diabetic feet through a nationwide big data investigation.

Recently, an epidemiological analysis was conducted in Korea using nationwide big data related to atraumatic lower extremity amputation caused by the aggravation of diabetic foot wounds [10,11,12]. Studies on diabetic foot- and diabetes-related amputations conducted overseas using big data can be analyzed for each country’s regional characteristics and population structure and extended to establishing plans to provide health services, policy interventions to mediate regional inequity, and preventive measures.

Accordingly, using open data to predict and inform risk for the systematic prevention and education of the aggravation of diabetic foot is crucial [7]. The present authors hypothesized that changes in the air quality in Korea would affect the aggravation of diabetic foot, amputation, and vascular surgery through a systematic literature review of previous studies and articles.

This study aimed to analyze the association between the aggravation of diabetic foot and the air quality in all regions and the entire population of Korea by using big data on air quality.

This study examined the association between atmospheric particulates, such as SO_2_, NO_2_, CO, and O_3_, and the frequency of diabetic foot amputation or vascular intervention.

The ultimate purpose of this study was to provide a basis for establishing large-scale diabetic foot management policies and implementing risk notices by tracing the association between air quality and diabetic foot deterioration and wounds and categorizing the identified associations into regions.

## 2. Materials and Methods

### 2.1. Study Population

Data were collected for a comprehensive analysis from open sources. Patient information, constituting big data, was extracted from the customized database of the National Health Insurance. The study cohort comprised 347,543 patients with diabetes diagnosed with diabetes or diabetic neuropathy between 2011 and 2019. Exclusion criteria were applied to ensure the integrity of the study cohort. Patients were excluded if they met any of the following criteria: 1503 individuals whose eligibility could not be determined according to the inclusion criteria, 206,688 who were diagnosed with diabetes in 2002, 4701 with less than two diabetes diagnoses, 145,936 with no history of a diabetes diagnosis (ICD 8, E104/114/124/134/144, or G590/632/990) or diabetes medication prescription, and 432 with missing addresses. After applying these exclusion criteria, 13,353 patients who underwent diabetic foot amputation and treatment were included (Figure 1).

### 2.2. Air Pollution Data

Big data on air quality were obtained from the Air Korea portal and the National Institute of Environmental Sciences. Each dataset was collected and preprocessed to analyze the impact of air quality factors on diabetic foot prognoses and amputation.

Air quality factors analyzed included sulfur dioxide (SO_2_), nitrogen dioxide (NO_2_), carbon monoxide (CO), and ozone (O_3_).

The air quality variables were standardized before the analysis to account for differences in units and scale. All independent variables were treated as quantitative data to ensure consistency in the analysis.

### 2.3. Health Data

Patient information, including diabetic foot exacerbation events, was extracted from the National Health Insurance database. The onset of diabetic foot aggravation was operationally defined as the first day of the operation or procedure, and relevant analyses were conducted using specified procedure codes. Atraumatic lower limb amputations due to diabetic foot complications were classified into major and minor categories based on procedure codes.

The analysis utilized the Korean Standard Classification of Diseases (KCD) 8 code entered at the time of surgical procedures. Procedure codes for lower limb vascular procedures included M6597, M6605, M6612, M6613, M6620, M6632, and M6633. Codes for vascular operations of the lower extremities included O0161-O0171 and O1643-6. Atraumatic lower limb amputations were categorized into major (N0571, N0572, and N0573) and minor (N0574 and N0575) types (Table 1).

### 2.4. Statistical Analysis

The lag period, representing the time interval between changes in air quality and subsequent amputation events, was calculated for each air quality factor. The frequency of diabetic foot amputations in each region was analyzed using a distributed lag non-linear model (DLNM).

All statistical analyses were conducted using SAS (version 9.4, SAS Institute, Cary, NC, USA) and R (version 3.6.3, R Foundation for Statistical Computing, Vienna, Austria). The monthly mean of daily climatic factors was calculated and normalized to the annual average and standard deviation to analyze the frequency of all procedures. A seasonal and trend decomposition analysis of time series was performed using X11, Seasonal Extraction in Arima Time Series (seat), and STL (Loess) methods to assess the time trend and seasonality of the monthly frequency of diabetic foot deterioration. The lag effect was determined using distributed lag non-linear models (DLNMs) combined with the Quasi-Poisson generalized linear model to estimate the time required for each climatic and air quality factor to impact the exacerbation of diabetic foot. Lag days ranging from approximately 1 to 14 days were computed based on the correlation estimation between the risk of diabetic foot amputation and DTR (per 1-degree Celsius increase), relative humidity (per 1% increase), and air pollutants (per 1 standard deviation increase) using the DLNM. Air quality variables were standardized before the analysis due to differences in units and scale. All independent variables were treated as quantitative data, and the DLNM was further analyzed using the Poisson generalized non-linear model. The regional risk of diabetic foot amputation was predicted and mapped for 16 cities and provinces using the DLNM, Kor-maps, and T-map packages (version 10.6.0).

## 3. Results

### 3.1. Study Population

Among individuals with diabetic foot ulcers, the mean age is 59.94 years, with a standard deviation of 11.81 years. The gender distribution shows a notable difference, with 60.79% being male and 32.21% female. Dyslipidemia is the most prevalent underlying condition among this group, followed by cardiovascular disease, anxiety, cerebrovascular disease, and depression. The distribution of subjects across different regions reveals that Gyeonggi-do has the largest proportion, while Jeju-do has the smallest (Table 2).

### 3.2. Health Data

The study analyzed the annual frequency of lower limb vascular procedures, lower limb vascular operations, and atraumatic lower limb amputations in patients with diabetic foot up to 10 years before the study period. The frequency of all procedures demonstrated a steady increase from 2001 to 2019 (refer to Figure 2, Table A1). No statistically significant cycles were observed for the annual procedures.

The rate of increase in the frequency of lower limb vascular procedures was the highest, while the rate of increase in the frequency of lower limb vascular operations was the lowest. An annual periodicity of lower limb amputations was noted, with significant increases observed in March, July, and November. Additionally, the frequency of amputation operations was relatively low in February and September (Figure 3).

### 3.3. Statistical Analysis (Associations between Air Pollution and Diabetic Foot Ulcer)

#### 3.3.1. Between Air Quality and SO_2_

The DLNM analysis results of SO_2_ concentration and diabetic foot amputation by region were expressed as RR according to the increase in 1 standard deviation of SO_2_.

The highest RR was observed in the Gangwon region, and Incheon showed a significant increase in risk at lag effects of 5 days or more.

Contrarily, some lag days of Daejeon, Ulsan, and Jeollabuk-do reduce the risk (Figure 4). Detailed values are included in Table A2.

#### 3.3.2. Between Air Quality and CO

The results of the DLNM analysis of CO concentration and diabetic foot amputation by region were expressed as RR with an increase in 1 standard deviation of CO.

As with SO_2_, the highest significant RR was found in the Gangwon region (RR = 1.456, *p* < 0.05, lag = 12; Figure 5).

Lag days of 1–4 days and 9 to 13 days in Busan significantly increased the risk (Figure 4). Detailed values are included in Table A3.

#### 3.3.3. Between Air Quality and NO_2_

The DLNM analysis results of NO_2_ concentration and diabetic foot amputation by region were expressed as RR with an increase in 1 standard deviation of NO_2_.

NO_2_ concentration significantly increased risk in the capital region, including Seoul, Incheon, and Gyeonggi-do, and in Chungcheongnam-do, Daegu, Ulsan, Gyeongsangbuk-do, and Jeollabuk-do.

In particular, a high RR was observed in Chungcheongnam-do (RR = 1.569, lag = 12; RR = 1.550, lag = 13; Figure 6). Detailed values are included in Table A4.

#### 3.3.4. Between Air Quality and O_3_

The results of the DLNM analysis of O_3_ concentration and diabetic foot amputation by region were expressed as RR according to an increase in 1 standard deviation of O_3_.

The highest RR was observed in Jeju (RR = 1.148, *p* > 0.5, lag = 14). Seoul, Gyeonggi-do, and Chungcheongbuk-do showed an increase in RR on shorter lag days, while Gwangju, Gyeongsangbuk-do, and Jeollabuk-do showed an increase in RR on longer lag days.

In Busan and Ulsan, a trend of increasing RR over most lag days was observed (Figure 7). Detailed values are included in Table A5.

## 4. Discussion

Diabetic foot is aggravated by diabetic vasculopathy, diabetic neuropathy, worsening of wounds and ulcers related to diabetic vasculopathy and neuropathy, and foot infections [3,4]. Prevention and management are particularly essential because of the risk of amputation of the lower limbs in the case of the aggravation of diabetic foot [5]. Patients who undergo the amputation of lower extremities have low survival rates, require a long time to adapt to daily life, and incur high socioeconomic costs due to medical expenditure and disability. The number of diabetic foot amputations in Korea increased every year from 2011 to 2016, resulting in a 47% increase in socioeconomic costs over five years [13,14]. These results implicate a higher rate of increase in socioeconomic costs for patients with diabetic feet in Korea compared to other developed countries. Therefore, it is necessary to analyze the aggravating factors of patients with diabetic feet, conduct a systematic analysis to prevent the amputation of the lower extremities, and manage and prevent diabetic feet at the national level.

Health education on diabetic foot is always important regardless of when, but an analysis of contributing factors of amputations and prevention are crucial to deter the exacerbation of the disease from progressing to amputation. As part of such an effort, it is necessary to consider changes in air quality factors, such as SO_2_, NO_2_, CO, and O_3_, to manage patients at home, medical institutions, and hospitals.

Although the precise underlying pathways between air pollution and diabetes have not yet been analyzed, various hypotheses have been proposed. It has been reported that exposure to relatively small particles increases the systemic inflammatory response and is associated with the oxidative reaction, endoplasmic reticulum stress, impaired endothelial function, cardiac autonomic nervous system dysfunction, and mitochondrial dysfunction [15,16,17]. It has also been shown that exposure to air pollutants can cause epigenetic changes, which can lead to changes in indicators of clotting, inflammation, and endothelial function [18,19].

NOx refers to nitrogen oxides, a byproduct of the high-temperature combustion process, and is an indicator of traffic-related air pollution. CO, or black carbon, is a component of particulate matter formed during diesel fuel combustion and is an indicator of air pollution produced by internal combustion engines. Notably, it is an indicator of air pollution caused by diesel engines. Ground-level O_3_ is formed when NO_2_ and volatile organic compounds react with sunlight and heat. SO_2_ is a byproduct of fossil fuel combustion, primarily in industrial resources. These factors may be measured more accurately in models describing the source of air pollutants than in models analyzing a single source of pollutants for individual exposures resulting from potential interactions [20,21,22,23]. Therefore, even if the exact mechanism of air pollution inducement and exacerbation of diseases has not been completely proven, this correlation study between the aggravation of diabetic foot and air pollution through a big data analysis is very significant as a model for epidemiological evidence.

In this study, SO_2_, CO, NO_2_, and O_3_ were identified to have significant RR for the aggravation of diabetic foot.

The examination of each air quality factor and weight by region showed a significant increase in RR in Gangwon-do and Incheon for SO_2_, in Gangwon-do and Busan for CO, and in the capital area, including Seoul, Incheon, and Gyeonggi-do, and Chungcheongnam-do, Daegu, Ulsan, Gyeongsangbuk-do, and Jeollabuk-do for NO_2_. O_3_ was shown to increase RR in cases of shorter lag effects in Seoul, Gyeonggi-do, and Chungcheongbuk-do, and in the cases of more prolonged lag effects in Gwangju, Gyeongsangbuk-do, and Jeollabuk-do.

This study also found that various air quality factors appeared to be related to the aggravation of diabetic foot. However, the effects of various air quality factors on the aggravation of diabetic foot differed slightly depending on the characteristics of each region. It is determined that various factors, such as demographic characteristics, socioeconomic differences, number of vehicles per unit area, distribution of internal combustion engines, cumulative exposure to air pollutants, hospital bed distribution by region, traffic conditions, and level of education, affected the regional RRs for each air quality factor.

This study has several limitations. This study used a self-controlled model in conjunction with the self-controlled case series design to control individual-level confounding factors, such as education level, basic medications, and disease history. However, individual-level confounders cannot be directly adjusted in the model, and confounder bias may remain. Further investigation is required to control more environmental information and individual-level variables. In addition, the land area of Korea is around 100,210 km^2^, which is very small compared to other countries that study the correlation between atmospheric factors and diabetes, such as the United States and China. Consequently, it was difficult to identify any noticeable changes in air quality factors for each region in this study. Furthermore, although there is open-source data on various information, such as dew point, wind speed, and air pressure, in addition to air quality, no studies have analyzed the association between these factors and diabetes.

Lastly, there is a limitation in that the present study did not include clinically detailed factors, such as the underlying disease of each patient, or personalized data, such as lifelog data among open personal health records.

Nevertheless, this study analyzed the factors affecting the actual living environment through a residence-based analysis rather than one based on the location of medical institutions and attempted to analyze big data more effectively by applying various models of the aggravation of diabetic foot, including vascular intervention, instead of analyzing only the lower extremity amputations as in previous studies.

## 5. Conclusions

Based on the results of this study, significance was observed between the aggravation of diabetic foot and various air quality factors. However, the air quality factors with high RRs differed by region.

Considering different variables and regional characteristics of air quality factors, it is believed that the cornerstone for regionally customized diabetic foot management should be refined.

## Figures and Tables

**Figure 1 ijerph-21-00775-f001:**
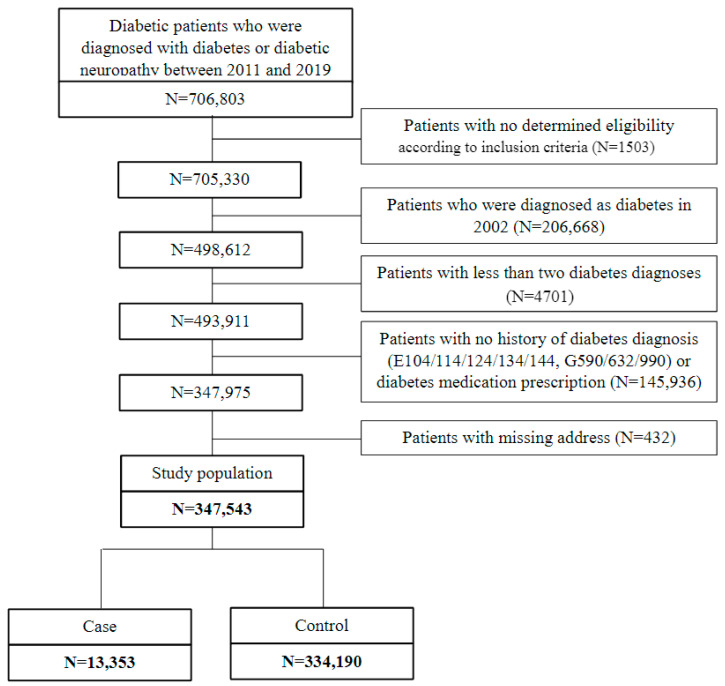
Schematics of subject selection. Note that “Case” means experimental group, and “Control” means control group.

**Figure 2 ijerph-21-00775-f002:**
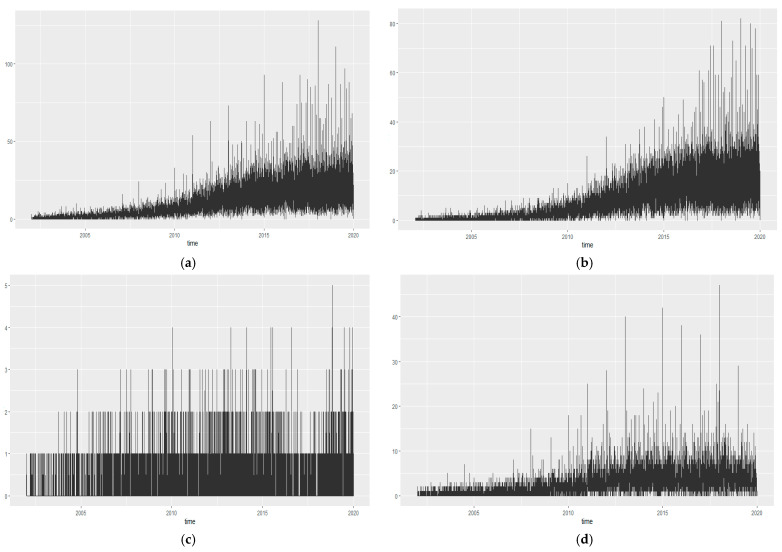
Frequencies of lower limb artery procedures, lower limb artery operations, and lower limb amputation in patients with diabetic foot. (**a**): Frequency of all procedures and operations; (**b**): Frequency of lower limb vascular procedures; (**c**): Frequency of lower limb vascular operations; (**d**): Frequency of lower limb amputations. Detailed values are presented in Appendix A. Y axis: number of cases performed.

**Figure 3 ijerph-21-00775-f003:**
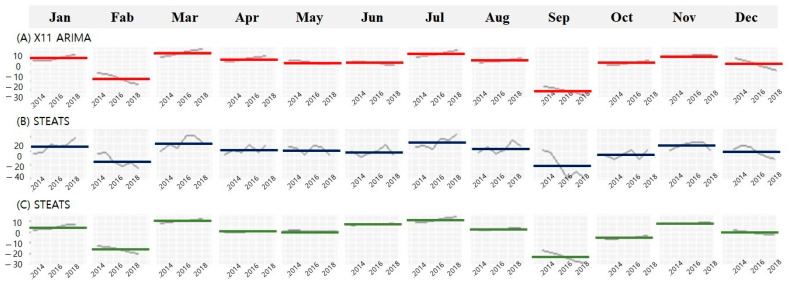
Trends in monthly lower limb amputation rates of patients with diabetic foot analyzed using three different time series decomposition methods. (**A**) X11-ARIMA: integrated autoregressive moving average processes. (**B**) SEATS: seasonal extraction in ARIMA time series. (**C**) STL: seasonal and trend decomposition using Loess methods. (Note) Black line: adjusted amputation rate by each method, blue line: mean of adjusted amputation rate by each method.

**Figure 4 ijerph-21-00775-f004:**
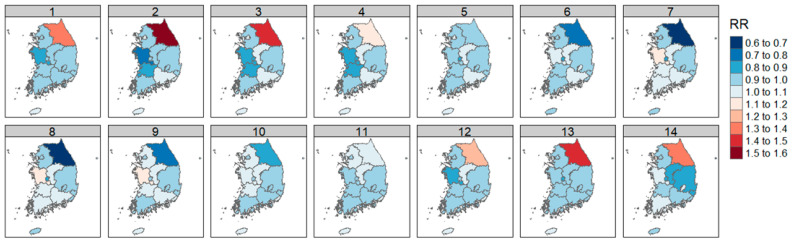
Correlation of diabetic foot amputation risk according to atmospheric sulfur dioxide concentration by region in Republic of Korea.

**Figure 5 ijerph-21-00775-f005:**
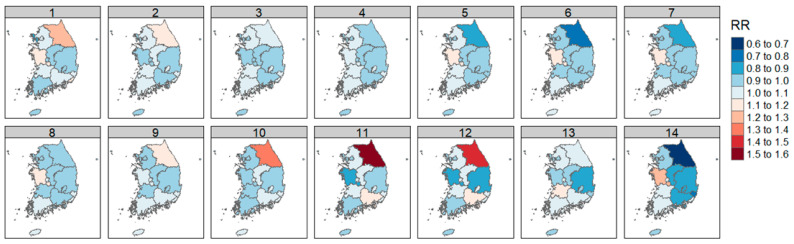
Correlation of diabetic foot amputation risk according to atmospheric carbon monoxide concentration by region in Republic of Korea.

**Figure 6 ijerph-21-00775-f006:**
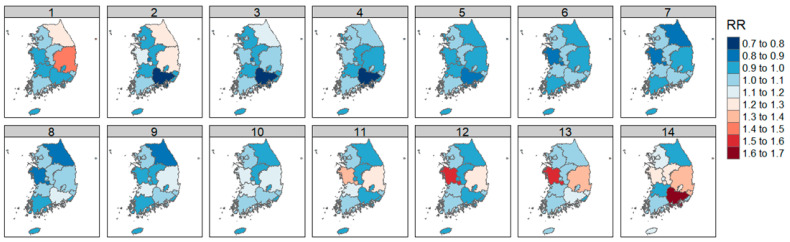
Correlation of diabetic foot amputation risk according to atmospheric nitrogen dioxide concentration by region in Republic of Korea.

**Figure 7 ijerph-21-00775-f007:**
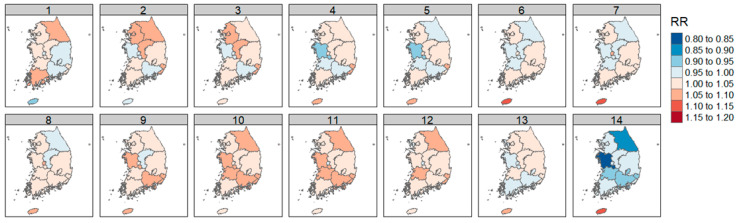
Correlation of diabetic foot amputation risk according to atmospheric ozone concentration by region in Republic of Korea.

**Table 1 ijerph-21-00775-t001:** The operational definition of the event of diabetic foot deterioration. Surgery codes for lower limb artery procedures and operations and lower extremity amputations due to aggravation of diabetic foot.

Category	Surgery Codes
A. Lower limb artery procedures	M6597, M6605, M6612, M6613, M6620, M6632, M6633
B. Lower limb artery operations	O0161-O0171, O1643-6
C. Lower limb amputation operation	N0571 to N0575 (major amputation: N0571, N0572, N0573; minor amputation: N0574, N0575)

**Table 2 ijerph-21-00775-t002:** Baseline characteristics of the study subjects.

Variable	Total	No Diabetic Foot Ulcer	Diabetic Foot Ulcer	*p*-Value
	(N = 347,543)	(N = 334,190)	(N = 13,353)	
Age (years)	61.69 ± 11.53	61.76 ± 11.52	59.94 ± 11.81	<0.001
Sex, n (%)				<0.001
Male	164,524 (47.34%)	155,472 (46.52%)	9052 (67.79%)	
Female	183,019 (52.66%)	178,718 (53.48%)	4301 (32.21%)	
Baseline comorbidities, n (%) †				
Hypertension	160,871 (46.29%)	155,781 (46.61%)	5090 (38.12%)	<0.001
Coronary artery disease	22,383 (6.44%)	21,445 (6.42%)	938 (7.02%)	0.005
Cerebrovascular disease	30,768 (8.85%)	29,623 (8.86%)	1145 (8.57%)	0.249
Region				<0.001
Seoul	56,423 (16.23%)	54,210 (16.22%)	2213 (16.57%)	
Incheon	13,521 (3.89%)	12,876 (3.85%)	645 (4.83%)	
Gyeong-gi	61,001 (17.55%)	58,559 (17.52%)	2442 (18.29%)	
Busan	27,437 (7.89%)	26,229 (7.85%)	1208 (9.05%)	
Daegu	13,617 (3.92%)	13,032 (3.9%)	585 (4.38%)	
Daejeon	8641 (2.49%)	8232 (2.46%)	409 (3.06%)	
Gwangju	7162 (2.06%)	6903 (2.07%)	259 (1.94%)	
Ulsan	6553 (1.89%)	6272 (1.88%)	281 (2.1%)	
Kyeongsangnam-do	27,378 (7.88%)	26,250 (7.85%)	1128 (8.45%)	
Kyeongsangbuk-do	31,871 (9.17%)	30,797 (9.22%)	1074 (8.04%)	
Jeollanam-do	21,589 (6.21%)	20,847 (6.24%)	742 (5.56%)	
Jeollabuk-do	17,622 (5.07%)	17,016 (5.09%)	606 (4.54%)	
Chungcheongnam-do	23,986 (6.90%)	23,294 (6.93%)	692 (5.18%)	
Chungcheongbuk-do	12,180 (3.5%)	11,726 (3.51%)	454 (3.4%)	
Gangwon-do	15,615 (4.49%)	15,120 (4.52%)	495 (3.71%)	
Jeju-do	2936 (0.84%)	2816 (0.84%)	120 (0.9%)	

Abbreviation: Data was reported as mean ± SD for continuous variables and n (%) for categorical variables. *p*-Value was computed by t-test for continuous variables and chi-square test or Fisher’s exact test for categorical variables as appropriate. † Number of samples with each comorbidity is given (ICD 10: Hypertension I10–16, Coronary artery disease I21–I25, Cerebrovascular disease I60, I61, I63, I64, I69).

## Data Availability

Due to the extensive volume of original data, it was not feasible to include it within this manuscript. Nevertheless, the data is available upon request via email. Further inquiries can be directed to the corresponding author.

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
