# Peer review of "The Risk of the Aggravation of Diabetic Foot According to Air Quality Factors in the Republic of Korea: A Nationwide Population-Based Study"

_ijerph, 2024, doi:10.3390/ijerph21060775_

Round 1

Reviewer 1 Report

Comments and Suggestions for Authors

This is a very interesting filed . I suggest to the authors to explain better their methods, in order to underline in the conclusions and discussion the link between air quality and diabetic foot  disease worsening. I also suggest to stratify, if possible , the Df population according to wifi classification. 

Comments on the Quality of English Language

 a moderate editing of english language is required

Author Response

Thank you for such kind review. We tried our best to clarify every section  as you can see it in the revised manuscript. About the wifi classification, it was impossible because the big data extracted from the national health insurance didn't provide any stratified information about wifi classification.

Some part of the manuscript are not finished yet, due to reviewer 2's comment, which is taking longer time.  So please be generous with it. We will be finishing it as soon as possible.

Reviewer 2 Report

Comments and Suggestions for Authors

This study investigates the short-term (acute, sub-acute) effects of air pollution on the exacerbation of diabetic foot. They analyze the association between diabetic foot occurrence and changes in daily mean air pollutant concentrations, by using national open data from health insurance services and air quality data from the National Institute of Environmental Science.

This epidemiological study uses a case-crossover design and calculates the association between short-term changes in air quality and the occurrence of amputation operations for diabetic foot for different lag periods (ranging from 1-14 days). Using a distributed lag non-linear model, they analyze the frequency of diabetic foot amputation in different regions, identifying specific pollutants such as SO2, CO, NO2, and O3 and their respective relative risks.

I found the study topic both important and very interesting. While the study benefits from a large dataset with almost 350,000 observations (including cases and controls) and corresponding air pollution data for specific days and lag periods, the presentation of results from this dataset could be improved.

The authors may collaborate with air pollution specialists to improve their understanding of environmental variables.

Abstract

The total number of cases and control included in the study should be provided in the abstract.

More information on effect estimates (relative risks) and their 95% confidence interval should be added to the abstract.

Methods

The methods section may be divided into sub-parts and each of them should be explained in more detail:

-2.1.Study population

-2.2.Air pollution data

Detailed information on air pollution exposure assessment methods should be provided.

-2.3.Health data

-2.4.Statistical analysis

Further explanation is needed regarding the choice of the case-crossover design for assessing the association between diabetic foot ulcers and air pollution.

Short –term effects of air pollution (daily changes in air pollution and daily changes in health outcome)

Results

The results section should be divided into sub-parts (like the methods part), each providing findings for specific subsections.

-3.1.Study population

Please move Table 1 to the results section and provide all information on the study population in this subsection.

-3.2.Air pollution data

Descriptive statistics for measured air pollution levels should be included.

Correlation between air pollutants should be given as well.

-3.3.Health data

Information on the number of cases and frequency of diabetic foot ulcer should be provided here. The temporal trends (increase during the years) may be pointed out in this subsection.

Please refer to Figure 2 here and explain the results more.

-3.4.Statistical analysis (associations between air pollution and diabetic food ulcer)

It is stated in line 115 that the study performed analyzes for particulate matter (PM10 and PM2.5), while the results are not given. Please add results for particulate matter and discuss.

As diabetic foot ulcer is more common among men (based on Table1), stratified analyses by sex should be performed and effect-modification by sex should be discussed.

Please edit the titles of sub-sections for clarity.  

Line 152 – 3.4.1. Associations between SO2 and diabetic foot exacerbation

Line 159 - 3.4.2. Associations between CO and diabetic foot exacerbation

Line 166 - 3.4.3. Associations between NO2 and diabetic foot exacerbation

Line 174 - 3.4.4. Associations between O3 and diabetic foot exacerbation

Line XXX - 3.4.5. Associations between PM10 and diabetic foot exacerbation (please add results for PM10)

Line XXX - 3.4.6. Associations between PM2.5 and diabetic foot exacerbation (please add results for PM2.5)

Figure 1 – please define what “Case” and “Control” exactly mean? 

Figure 2 - Y axis label is not given, please add a label that defines the Y-axis variable. Maybe  daily frequency of diabetic foot ulcer cases?

Figure 3-6

Correlation and relative risk? I did not understand this.

Correlation between variables is usually given with correlation coefficient from 0 to 1. This may be something else correlation, maybe the association between diabetic foot and air pollution per region?

Figure 5 - NO2 is nitrogen dioxide. (not nitrate dioxide). Please correct.

Discussion

Comparison with findings of other relevant studies should be incorporated.

The lag period between changes in air quality and the occurrence of amputation operations for diabetic foot should be discussed more.

Line 228- Clarification is needed regarding the distinction between gases and particulate air pollution (CO is not a component of particulate matter, it is a gas)

Apprendix

Appendix A- As the outcome types 1, 2, and 3 are mentioned only in Korean, clarification is needed here.

A list of abbreviations should be provided, for instance DLNM,  ICD, KCD, RR, SO2, CO, NO2, O3, PM, … etc.

Comments on the Quality of English Language

Minor editing of English language required

Author Response

Thank you for such detailed review. While revising the manuscript according to your review, our paper has become much more intuitive and strength in depth.

I will reply to your every single comment below.

------------------------------------------------------------------------------------

Abstract

The total number of cases and control included in the study should be provided in the abstract.

-> done in the manuscript

More information on effect estimates (relative risks) and their 95% confidence interval should be added to the abstract.

-> done in the manuscript

Methods

The methods section may be divided into sub-parts and each of them should be explained in more detail:

-2.1.Study population

-> done in the manuscript

-2.2.Air pollution data

Detailed information on air pollution exposure assessment methods should be provided.

-> done in the manuscript

-2.3.Health data

-> done in the manuscript

-2.4.Statistical analysis

Further explanation is needed regarding the choice of the case-crossover design for assessing the association between diabetic foot ulcers and air pollution.

Short –term effects of air pollution (daily changes in air pollution and daily changes in health outcome)

-> done in the manuscript

Results

The results section should be divided into sub-parts (like the methods part), each providing findings for specific subsections.

-3.1.Study population

Please move Table 1 to the results section and provide all information on the study population in this subsection.

-> done in the manuscript

-3.2.Air pollution data

Descriptive statistics for measured air pollution levels should be included.

Correlation between air pollutants should be given as well.

-> The authors truly think this is in need and the authors are working on it. Please allow the authors some more time, the authors will finish it as soon as possible. 

-3.3.Health data

Information on the number of cases and frequency of diabetic foot ulcer should be provided here. The temporal trends (increase during the years) may be pointed out in this subsection.

-> done in the manuscript

Please refer to Figure 2 here and explain the results more.

-> done in the manuscript

-3.4.Statistical analysis (associations between air pollution and diabetic food ulcer)

It is stated in line 115 that the study performed analyzes for particulate matter (PM10 and PM2.5), while the results are not given. Please add results for particulate matter and discuss.

-> The authors conducted PM10 and PM2.5 in another study of ours. It was an accident to mention those in this manuscript, and the authors deleted it.

As diabetic foot ulcer is more common among men (based on Table1), stratified analyses by sex should be performed and effect-modification by sex should be discussed.

-> This was something the authors couldn't come up with and the authors are truly thankful for pointing this out. The authors are now on progress so please allow us some more time. Thank you.

Please edit the titles of sub-sections for clarity.  

Line 152 – 3.4.1. Associations between SO2 and diabetic foot exacerbation

-> done in the manuscript

Line 159 - 3.4.2. Associations between CO and diabetic foot exacerbation

-> done in the manuscript

Line 166 - 3.4.3. Associations between NO2 and diabetic foot exacerbation

-> done in the manuscript

Line 174 - 3.4.4. Associations between O3 and diabetic foot exacerbation

-> done in the manuscript

Line XXX - 3.4.5. Associations between PM10 and diabetic foot exacerbation (please add results for PM10)

-> deleted

Line XXX - 3.4.6. Associations between PM2.5 and diabetic foot exacerbation (please add results for PM2.5)

-> deleted

Figure 1 – please define what “Case” and “Control” exactly mean?

-> done in the manuscript

Figure 2 - Y axis label is not given, please add a label that defines the Y-axis variable. Maybe  daily frequency of diabetic foot ulcer cases?

-> done in the manuscript

Figure 3-6

Correlation and relative risk? I did not understand this.

Correlation between variables is usually given with correlation coefficient from 0 to 1. This may be something else correlation, maybe the association between diabetic foot and air pollution per region?

-> yet finished, working on this part as well. 

Figure 5 - NO2 is nitrogen dioxide. (not nitrate dioxide). Please correct.

-> done in the manuscript

Discussion

Comparison with findings of other relevant studies should be incorporated.

The lag period between changes in air quality and the occurrence of amputation operations for diabetic foot should be discussed more.

Line 228- Clarification is needed regarding the distinction between gases and particulate air pollution (CO is not a component of particulate matter, it is a gas)

-> Thank you for the comment, the authors are now working on the discussion section. Please allow the authors some more time.

Apprendix

Appendix A- As the outcome types 1, 2, and 3 are mentioned only in Korean, clarification is needed here.

-> done in the manuscript

A list of abbreviations should be provided, for instance DLNM,  ICD, KCD, RR, SO2, CO, NO2, O3, PM, … etc.

-> done in the manuscript

-----------------------------------------------------------------------------------

Your comments gave the authors the opportunity to take a completely new look at this study from a reader's perspective. Perhaps the authors were caught up in narrow-minded thinking while conducting this research. The authors plan to complete the review of the discussion section and other unfinished parts as soon as possible, so please give the authors more time.

Thank you.
